# Does the Epstein–Barr Virus Play a Role in the Pathogenesis of Graves’ Disease?

**DOI:** 10.3390/ijms20133145

**Published:** 2019-06-27

**Authors:** Aleksandra Pyzik, Ewelina Grywalska, Beata Matyjaszek-Matuszek, Jarosław Ludian, Ewa Kiszczak-Bochyńska, Agata Smoleń, Jacek Roliński, Dawid Pyzik

**Affiliations:** 1Department of Clinical Immunology, Medical University of Lublin, 20-093 Lublin, Poland; 2Department of Clinical Immunology, Center of Oncology of the Lublin Region St. Jana z Dukli, 20-090 Lublin, Poland; 3Department of Endocrinology, Medical University of Lublin, 20-954 Lublin, Poland; 4Department of Epidemiology, Medical University of Lublin, 20-080 Lublin, Poland; 5Department of Trauma and Orthopedic Surgery, Center of Oncology in Lublin, 20-090 Lublin, Poland

**Keywords:** Graves’ disease, viruses, EBV, hyperthyroidism, autoimmune thyroid disease

## Abstract

Graves’ disease (GD) it the most common chronic organ-specific thyroid disorder without a fully recognized etiology. The pathogenesis of the disease accounts for an interaction between genetic, environmental, and immunological factors. The most important environmental factors include viral and bacterial infections. The Epstein-Barr virus (EBV) is one of the most common latent human viruses. Literature has suggested its role in the development of certain allergic and autoimmune diseases. EBV also exhibits oncogenic properties. The aim of the study was to analyze and compare the presence of EBV DNA in peripheral blood mononuclear cells (PBMCs) in patients with newly recognized GD and to find a correlation between EBV infection and the clinical picture of GD. The study included 39 untreated patients with newly diagnosed GD and a control group of 20 healthy volunteers who were gender and age matched. EBV DNA was detected with reverse transcription polymerase chain reaction (RT PCR) assay. The studies showed a significantly higher incidence of EBV copies in PBMCs among GD patients compared to the control group. Whereas, no significant correlations were found between the incidence of EBV copies and the evaluated clinical parameters. Our results suggest a probable role of EBV in GD development. EBV infection does not affect the clinical picture of Graves’ disease.

## 1. Introduction

### 1.1. Graves’ Disease

Graves’ disease (GD) belongs to a heterogeneous group of chronic organ-specific thyroid disorders without a fully recognized etiology [1]. It is the most common cause of hyperthyroidism in iodine-sufficient areas [2]. In developed countries, it affects approximately 0.5% of the general population.

Patients suffer from a vascular goiter and often present extrathyroidal symptoms, such as orbitopathy (~50% of the cases) or more rarely dermopathy or acropachy. Characteristic somatic symptoms of hyperthyroidism include the following: Heart palpitation, weight loss, diarrhea, hand tremors, sweating, muscle weakness, and menstrual disorders, whereas neuropsychiatric symptoms include irritability, insomnia, fatigue, or anxiety [3,4]. The disease is characterized by the presence of specific IgG antibodies, mainly against TSH receptors (thyroid stimulating antibody), which bind with the receptor on the surface of thyroid follicular cells and stimulate their growth. Other substances, like thyroglobulin (Tg), thyroperoxidase (TPO), and sodium-iodide symporter, are also responsible for autoimmunization of lymphocytes [3].

The pathogenesis of GD is shown in Figure 1.

The most important environmental factors include viral (human herpesvirus 7, Kaposi’s sarcoma-associated herpesvirus, hepatitis C virus, retroviruses, and influenza B virus) and bacterial (*Yersinia enterocolitica*, *Borrelia burgdorferi*, *Helicobacter pylori*) infections, where the stimulation of the autoimmune response develops through the modification of the patient’s own antigens, molecular mimicry, induction of T-cell response, and increased expression of human leukocyte antigen (HLA) system molecules on thyrocytes [12,13,14].

Recent years have seen increasing interest in latent viruses. One of the most prevalent is the Epstein–Barr virus (EBV), which can modulate the immune response, e.g., through encoding antiapoptotic Bcl-2 homolog (BHRF-1).

### 1.2. EBV

EBV, also classified as human herpesvirus 4, belongs to the *Herpesviridae* family, whose only natural host is man.

It is estimated that almost 95% of the adult population are carriers of this virus, whereas in developing countries it may be as much as 100% [15,16]. The primary infection is most often transmitted through saliva (the so-called “kissing disease”) in children and adolescents. The virus can also be transferred by blood or with a transplanted organ. Its life cycle includes two phases: Lytic and latent [15]. In most people, it takes the latent form in B lymphocytes and in epithelial cells of the nasopharynx, although it may also infect T and NK cells. It replicates in the nucleus where it is suppressed with normal T cells and shows no clinical symptoms. Sometimes, especially in young adults, the infection can be symptomatic (fever, joint and muscle pain, sore throat, rash, diarrhea, hepato- and splenomegaly, lymphadenopathy [17]). The virus may reactivate in favorable conditions, such as immunosuppression. Within just a few hours of infection, the cell undergoes a number of changes, such as deregulation of the cell cycle, stimulation of growth factors, inhibition of apoptosis, and immunomodulation, which lead to “cell immortality”. After a few days, the DNA of the cell may replicate with the DNA of the virus in an uncontrolled way [18].

In the diagnosis of EBV infection, simple non-specific screening tests are employed to detect heterophile antibodies, which are dependent on the agglutination of sheep, horse, or bovine erythrocytes (the Paul–Bunnell–Davidsohn test; the Monospot test). Such tests may lead to incorrect diagnosis of mononucleosis in the course of other infectious diseases due to the fact that heterophile antibodies persist in blood for a long time. There can also be false-positive results due to low levels of antibodies [19]. Specific serological tests (ELISA, Western-blot) detect the presence and number of specific antibodies against numerous viral antigens, such as: Epstein–Barr nuclear antigen-1 (EBNA-1), Epstein–Barr nuclear antigen-2 (EBNA-2), Epstein–Barr nuclear antigen-leader protein (EBNA-LP), latent membrane protein-1 (LMP-1), EBNA-3A, EBNA-3B, and EBNA-3C. Unfortunately, such tests are not perfect and do not directly reflect the status of EBV, and the high levels of some antibodies may remain until death, and therefore results are difficult to interpret. Antibody concentrations do not correspond to the gravity of the disease or its duration. Much more reliable results are given by molecular methods, such as PCR and in situ hybridization [15,17]. They are highly recommended in the following cases: Infections of neonates and infants, in patients undergoing immunosuppression, and in diagnosing proliferative diseases of B lymphocytes. This method is also preferred in atypical forms of infections, such as in post-transplant lymphoproliferative disorder, in chronic active EBV infection, and in hemophagocytic lymphohistiocytosis [19].

## 2. Aim of the Study

The aim of the study was to analyze and compare the presence of genetic material in the form of EBV DNA in peripheral blood mononuclear cells (PBMCs) and to detect the number of DNA copies of this pathogen in patients with newly diagnosed untreated GD as well as to find a correlation between EBV infection and the clinical picture of GD. We hypothesized that the presence of an exponent of the reactivation of Epstein–Barr virus infection in the form of the presence of EBV DNA in the peripheral blood is associated with the specific severity of GD symptoms and some laboratory tests (levels of thyroid-stimulating hormone (TSH), free triiodothyronine (FT3), free thyroxine (FT4), thyroid-stimulating immunoglobulin (TSI), thyroperoxidase antibodies (anti-TPO), anti-thyroglobulin (anti-TG), leukocytes, and lymphocytes).

## 3. Results

The study showed a significantly higher incidence of EBV copies in PBMCs among GD patients compared with the control group. EBV DNA was not detected in healthy controls.

The presence EBV DNA was more frequent in women, but the number of EBV copies was higher in men. A comparison of the presence and number of copies of EBV DNA in PBMCs between the groups is shown in Table 1.

No significant correlations between the incidence of EBV DNA in the study group and clinical neuropsychiatric symptoms and signs, such as irritability, emotional lability, sleep disorders, and fatigue, or somatic manifestations of hyperthyroidism, such as weight loss, heart palpitation, heat intolerance, excessive sweating, menstrual disorder, muscle weakness, orbitopathy, goiter, tachycardia, velvet skin, muscle trembling, superficial tendon reflexes, high amplitude of blood pressure, pretibial myxedema, and thyroid acropachy, were found (Table 2).

As many as 11 of the 12 GD patients with EBV DNA in blood serum complained of chronic fatigue; however, this fact was non-significant.

No significant correlations between the incidence of EBV DNA in the study group and evaluated laboratory parameters, such as the level of TSH, FT3, FT4, TSI, anti-TPO, and anti-TG, were found (Table 3).

The absolute number of lymphocytes was significantly lower in the study group than in the control group. The percentage of TCD4 and lymphocytes in patients with GD was statistically significantly higher than in healthy subjects (*p* < 0.001), while the absolute number of these cells was significantly lower in this group (0.04). In the study group, the percentage and number of CD8 and lymphocytes were significantly lower compared to healthy subjects (*p* < 0.001) (Table 4).

## 4. Discussion

GD is an autoimmune thyroid disease, which corresponds to 60% to 80% of all cases of hyperthyroidism and is characterized by a variable course and frequent relapses in 40% to 60% of cases [20]. The aim of this study was to evaluate the correlation between EBV infection and GD development.

According to Toussirot, EBV meets all the criteria of an autoimmune causal factor: Prevalence, chronic latent infection, and the ability to reactivate and manipulate the immune response of its host to its own benefit [21]. EBV has developed several mechanisms that enable it to escape the immune system radar in its latent phase. These are the ability to decrease the number of exposed antigens, to encode proteins inhibiting the apoptosis of B lymphocytes in which the virus replicates (including viral homologue of the Bcl-2 apoptosis inhibitor), and to disturb the cytokine microenvironment [17]. Increased concentrations of interleukin 1 (IL-1), IL-6, and tumor necrosis factor result in an increased expression of HLA class II, which facilitates the presentation of its own antigens. A decreased concentration of interferon gamma weakens the antiviral response, while a high level of IL-10 increases the expression of Bcl-2 antiapoptotic protein [22]. In vitro experiments have shown that infecting B lymphocytes with the EBV of healthy people leads to their proliferation and production of monoclonal reactive autoantibodies that show an affinity to antigens of various tissues and organs. Such cells start to produce antiapoptotic particles encoded by the virus and locate in human organs as latently infected memory B cells. Then, they take the function of cells presenting antigens. In cross reactivity, T helper cells are activated, which, thanks to additional signals from the infected B cells, do not undergo apoptosis, but instead proliferate and infiltrate tissues. Cytokines produced by T lymphocytes increase the hormonal response. Highly specific IgG against thyroid antigens as well as polyreactive antibodies with a low affinity targeted against Tg and TPO are produced [23]. Afterwards, thyroid-stimulating immunoglobulin (TSI) antibodies bind with thyroid-stimulating hormone receptor (TSHR) in the thyroid and result in the proliferation of thyrocytes and the development of hyperthyroidism.

After some time, the healthy immune system destroys infected B lymphocytes through the mechanism of cytotoxicity with cytotoxic T cells. However, cells that were latently infected and B memory cells that do not proliferate and are not destroyed remain dormant in blood at various levels throughout the patient’s life. Muting immune system reactions results in disease remission. However, any further contact with the antigen may trigger the virus to reactivate and switch to the lytic phase of replication. Clinically, we can observe disease exacerbation [24].

Our study showed significantly lower lymphocytosis in peripheral blood in the study group (*p* = 0.02), in particular a significant drop in the percentage and number of CD8+ lymphocytes (*p* < 0.001). Such a phenomenon is in line with the available literature, where we can find a correlation between lymphopenia and the development of autoimmune diseases, such as systemic lupus erythematosus and rheumatoid arthritis. One of the hypotheses assumes that an insufficient number of cytotoxic cells weakens the defense against intracellular pathogens, especially latent viruses (EBV, HSV, cytomegalovirus, and others). The underlying cause of CD8+ deficiency is probably genetic due to the fact the such a phenomenon is present in healthy relatives of the patients suffering from an autoimmune disease [25], and its effect is the proliferation of those microorganisms in infected cells.

In 2008, Thomas et al. observed a significantly higher level of anti-EBV IgG antibodies in a study group than in a control group and formulated a hypothesis that EBV infection might play a role in the pathogenesis of autoimmune thyroid disease in children [23]. In later years, Akahori et al. further investigated this phenomenon and showed three cases of women with newly diagnosed GD and acute primary EBV infection in the form of infectious mononucleosis [26]. Nonetheless, other studies revealed that the level of early antigen antibodies, which is indicative of the reactivation of the virus, shows a significant correlation with the level of anti-TSHR in GD patients [27].

Some studies determined the presence of EBV-encoded RNA (EBER) in the cells of thyroid follicular epithelium. Janegova et al. employed the same method and presented a high prevalence of EBV infection in both autoimmune thyroid diseases (in GD, 62.5% of all cases; in Hashimoto’s disease, 80.7% of all cases) [28]. In the same year, and with the use of nuclear RNA fragments, Nagata et al. showed that EBV reactivation in TRAb(+) EBV(+) cells represented by EBER1(+) cells stimulates the host’s CD19+ B lymphocytes to increase the production of thyroid-stimulating antibodies, which may cause the development and exacerbation of GD [29]. Additionally, the authors noted that chronic inflammation in autoimmune diseases may be linked to the development of cancer, because as many as 90% of all primary thyroid lymphomas develop in patients suffering from Hashimoto’s disease [28].

A thorough review of the literature to date suggests there are no available data on studies evaluating the presence and number of copies of EBV DNA in GD, which gives the most reliable data on the activity of the infection in real time. Basic diagnostics of EBV infection are based on serological methods, which are not perfectly reliable. They do not directly reflect EBV status, and high levels of some antibodies may persist for many months, or sometimes even throughout the patient’s life, and are difficult to interpret. Moreover, antibody concentrations do not correspond to the gravity of the disease or its duration. To the best of our knowledge, our study is the first to evaluate DNA in PBMCs in GD patients. Such tests have higher sensitivity and specificity than serological ones because they allow the evaluation of the intensification of viremia at the time of the occurrence of hyperthyroidism symptoms.

Our study proved the high prevalence of the copies of EBV in PBMCs in GD patients as compared to the control group, in which no genetic material of the virus was detected (*p* = 0.01, χ² = 5.94). So far, a higher amount of EBV DNA in peripheral blood was found in the patients suffering from other autoaggressive diseases, such systemic lupus erythematosus [30], rheumatoid arthritis [31], primary cirrhosis of the liver [32], or multiple sclerosis [33].

Our earlier published studies on the expression of the particle of the receptor of the programmed cell death protein-1 (PD-1) serve as an additional confirmation of the role of viral infections in the pathogenesis of GD. This protein belongs to the group of negative regulators of the immune response. The mechanism of PD-1 action involves muting the inflammatory reaction by inhibition of cytolysis and activation of apoptosis. This leads to the transition of activated lymphocytes into anergic cells—the so-called “exhausted lymphocytes”—which results in blocking cytokine production and resetting tolerance. However, continuing high expression of PD-1 particles on T cells by inhibition of the production of interferon leads to chronic inflammation. The most important factors stimulating PD-1 expression on T cells include viral infections (hepatitis B virus, polyomaviridae, EBV, cytomegalovirus, and varicella zoster virus). Our earlier studies showed a significantly higher number, absolute value, and mean fluorescence intensity of CD4+, CD8+ T cells, and CD19+ B cells showing PD-1 expression in GD patients as compared to healthy volunteers, which may corroborate the above hypothesis [34].

EBV attacks various organs, most often the lymph nodes, liver, and spleen, but in most cases, the infection is asymptomatic or displays non-characteristic symptoms. In the clinical picture of GD patients, nonspecific symptoms dominated and included fatigue, weight loss, irritability, and heart palpitations. Among the typical organ lesions in GD patients, the most frequent symptom was vascular goiter, especially in women.

Our studies showed no significant correlations between the incidence of EBV and reported neuropsychiatric symptoms or signs, such as irritability, emotional lability, sleep disorders, and fatigue, and somatic manifestations of hyperthyroidism, such as weight loss, heart palpitation, heat intolerance, excessive sweating, menstrual disorder, muscle weakness, orbitopathy, goiter, tachycardia, velvet skin, muscle trembling, superficial tendon reflexes, high amplitude of blood pressure, pretibial myxedema, and thyroid acropachy.

The small sample size can be recognized as the main limitation of our study. Therefore, we intend to conduct further studies, in particular to analyze associations between the level of EBV viremia and the concentration of thyroid hormones, thyrotropic hormone, or antithyroid antibodies. It is important to find early immune or thyroid parameters that predispose a patient to GD development, in order to prevent clinically overt hyperthyroidism.

## 5. Material and Methods

The study included 39 untreated patients (32 women and 7 men) with newly diagnosed hyperthyroidism in the course of GD, aged 22 to 95 years (mean age 41.49 ± 15.74 years; median 39 years), under the care of the Endocrinology Clinic of the Medical University of Lublin in the years 2016–2018. The diagnosis was based on the positive result of antibodies against TSHR, while the clinical activity was assessed on the basis of the patient’s medical history, physical examination, and hormone tests (TSH, FT4, and FT3).

The control group consisted of 20 healthy volunteers (15 women and 5 men) with an average age, similar to the study group, of 42.15 ± 10.38 years (min. 29 years; max. 60 years; median 40 years). The general characteristics of the study and the control group are shown in Table 5. The patients treated with immunosuppressive agents, hormonal therapy, and those who had undergone blood transfusion, as well as the patients with a positive history of other autoimmune disorders, suffering from allergies, or those who had shown some symptoms of infection in the last two months were excluded from the study.

Neither the patients nor the controls used immunomodulating agents, vaccines, or hormonal preparations; showed signs of infection within at least six months prior to the study; underwent blood transfusion; or presented with an allergy. Moreover, none of the patients and controls had a history of oncological therapy or prior treatment for an autoimmune condition or tuberculosis or other chronic conditions that could be associated with impaired cellular or humoral immunity.

The blood was collected from the antecubital vein in the total amount of 20 mL (10 mL to a syringe with heparin, 5 mL to a test tube with ethylenediaminetetraacetic acid (EDTA), and 5 mL to a clot test tube; aspiration and vacuum systems by Sarstedt, Nümbrecht, Germany) in the course of routine biochemical tests. Mononuclear cells were immediately isolated from the collected blood, and plasma and serum were obtained to perform further assays.

Blood from the EDTA-coated tubes was used to isolate peripheral blood mononuclear cells (PBMCs) by density gradient centrifugation. Briefly, we layered 5 mL of whole blood diluted with 5 mL of normal saline on 5 mL of Ficoll-Paque™ (Miltenyi Biotec, Bergisch Gladbach, Germany) in 15 mm tubes. The tubes were centrifuged at 400 *g* for 30 min without brake. PBMCs were removed from buffy coats with a Pasteur pipette. The Blood-Serum was frozen, we did not use it in this study.

Following DNA isolation with a QIAamp DNA Blood Mini Kit (QIAGEN, Hilden, Germany), the number of EBV copies in PBMCs was detected using real-time PCR assay by means of the ISEX variant of the EBV PCR kit (GeneProof, Brno, Czech Republic), as described previously [35]. The sensitivity of the test was established at 10 DNA copies per 1 µL. The results are expressed in units of EBV DNA copies/µg of DNA, EBV DNA copies/mL of PBMCs suspension, and EBV DNA copies/100,000 cells.

The statistical analysis was performed with StatSoft, Poland Statistica v. 10.0 software. Categorical variables are expressed as the number of observations and percentage, whereas continuous variables are shown as means, standard deviations, medians, and first and third quartiles. Because the distributions of data were skewed (assessed with Shapiro–Wilk W test) or variances were heterogenous (assessed with Fisher’s F test), nonparametric statistics were applied to analyze differences between groups. For the comparison of two groups, Mann–Whitney’s U test was used, and for comparison of three or more subgroups, the Kruskal–Wallis H test followed by multiple-comparison post-hoc test was used. The Spearman R coefficient was used to analyze correlations between variables. For the analysis of categorical variables with a small number of observations (*n* < 5), the χ^2^ test with Yates correction was applied. Furthermore, *p* values less than 0.05 were considered significant.

## 6. Conclusions

A significantly higher presence of EBV DNA copies in PBMCs in patients newly diagnosed with GD as compared to the control group suggests a probable role of the virus in the development of GD. The presence of EBV DNA has no effect on the severity of hyperthyroidism, both in laboratory parameters and clinical course.

## Figures and Tables

**Figure 1 ijms-20-03145-f001:**
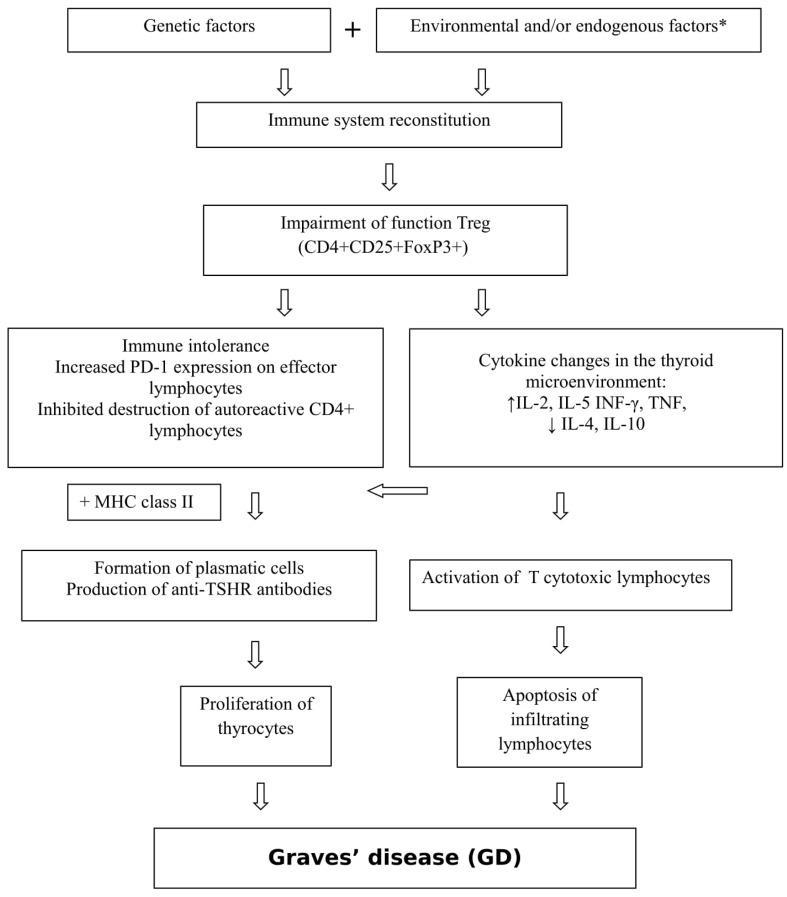
Pathogenesis of Graves’ disease (GD). * viral and bacterial infections, iodine, stress, medications, external radiation, radioactive iodine, toxins, selenium and vitamin D3 deficiency, lymphopenia, smoking, CD8+ T lymphocyte deficiency [5,6,7,8,9,10,11].

**Table 1 ijms-20-03145-t001:** Comparison of the presence and number of EBV DNA copies in PBMCs between the groups.

Parameter			Study Group (39)		Control Group (20)	*p*	χ²
**EBV DNA Present Number (%)**		**Men**	2	12 (30.77%)	0 (0%)	0.01	5.94
		**women**	10				
Number of EBV DNA copies /mL	men	Median (min.–max.)	4874.5 (600.8–9148.21)				
		(Q1–Q3)	2737.65–7011.36				
	women	Median (min.–max.)	1681 (620.44–27,339.30)				
		(Q1–Q3)	676.43–4202.35				
Number of EBV DNA copies (copies of EBV DNA/μgDNA)	men	Median (min.–max.)	41.62 (22.35–60.89)				
		(Q1–Q3)	31.97–51.26				
	women	Median (min.–max.)	29.6 (9.27–659.1)				
		(Q1–Q3)	11.99–136.23				
Number of EBV DNA copies (copies of EBV DNA/100,000 cells)	men	Median (min.–max.)	27.47 (14.75–40.19)				
		(Q1–Q3)	21.11–33.83				
	women	Median (min.–max.)	19.53 (6.12–435)				
		(Q1–Q3)	7.91–89.92				

Q1, first quartile; Q3, third quartile.

**Table 2 ijms-20-03145-t002:** Correlations between clinical neuropsychiatric and somatic manifestations of hyperthyroidism in patients in the study group with the presence of EBV DNA in peripheral blood.

Symptoms and Signs	EBV DNA (+)	EBV DNA (−)	*p*	χ²
**Neuropsychiatric**				
Irritability	8 (20.51%)	16 (41.03%)	0.66	0.19*
Emotional lability	4 (10.26%)	12 (30.77%)	0.77	0.09 **
Sleep disorders	9 (23.08%)	17 (43.59%)	0.46	0.54 *
Fatigue	11 (28.21%)	22 (56.41%)	0.42	0.66 *
**Somatic**				
Weight loss	9 (23.08%)	20 (51.28%)	0.95	0.004 *
Heart palpitation	9 (23.08%)	19 (48.72%)	0.77	0.88 *
Heat intolerance	7 (17.95%)	14 (35.90%)	0.71	0.14 *
Excessive sweating	7 (17.95%)	15 (38.46%)	0.87	0.03 *
Menstrual disorder	0 (0)	3 (9.38%)	0.22	1.50**
Muscle weakness	7 (17.95%)	13 (33.33%)	0.56	0.34 *
Orbitopathy	2 (5.13%)	3 (7.69%)	0.63	0.23 **
Goiter	12 (30.77%)	27 (69.23%)	0.77	1.13 *
Tachycardia	6 (15.38%)	19 (48.72%)	0.22	1.50 *
Velvet skin	9 (23.08%)	23 (58.97%)	0.44	0.59 *
Muscle trembling	8 (20.51%)	16 (41.03%)	0.66	0.19 *
Superficial tendon reflexes	1 (2.56%)	3 (7.69%)	0.79	0.07 **
High amplitude of blood pressure	2 (5.13%)	4 (10.26%)	0.88	0.02 **
Pretibial myxedema	1 (2.56%)	0	-	-
Thyroid acropachy	1 (2.56%)	0	-	-

* Pearson’s Chi^2^, ** Chi^2^ with Yates correction.

**Table 3 ijms-20-03145-t003:** Correlation between the presence of EBV DNA in the patients in the study group and laboratory parameters.

Parameter			Present EBV DNA	*p*	z
**TSI (U/L)**	**value**	Median (min–max)	11.95 (2.20–38.50)	0.68	–0.41
		Q1–Q3	5.15–20.75		
Anti-TPO (U/mL)		Median (min–max)	774 (29.00–3000.00)	0.84	0.20
		Q1–Q3	120.36–2464.15		
Anti-TG (IU/mL)		Median (min–max)	109.42 (10.00–407.00)	0.41	0.83
		Q1–Q3	15.00–231.50		
TSH (mIU/L)		Median (min–max)	0.008 (0.005–0.008)	0.82	0.36
		Q1–Q3	0.008–0.008		
FT4 (ng/dL)		Median (min–max)	3.94 (2.23–5.08)	0.16	1.39
		Q1–Q3	3.00–4.59		
FT3 (pg/mL)		Median (min–max)	14.10 (5.60–20.00)	0.28	1.10
		Q1–Q3	9.30–17.60		

Q1, first quartile; Q3, third quartile.

**Table 4 ijms-20-03145-t004:** Distribution of lymphocytes in the study and the control groups.

Parameter		Study Group (39)	Control Group (20)	*p* Value
**Lymphocytes (1 × 10^9^/L)**	Mean ± SD	2.01 ± 0.66	2.35 ± 0.59	0.02
	Median (min–max)	1.79 (1.25–4.18)	2.36 (1.39–3.38)	
CD4+ (%)	Mean ± SD	49.64 ± 7.5	44.46 ± 2.50	<0.001
	Median (min–max)	48.67 (22.85–62.63)	44.16 (40.71–48.84)	
CD4+ (10^3^/mm^3^)	Mean ± SD	0.89 ± 0.35	1.04 ± 0.27	0.04
	Median (min–max)	0.87 (0.59–2.44)	1.04 (0.62–1.54)	
CD8+ (%)	Mean ± SD	26.95 ± 4.28	34.36 ± 3.29	<0.001
	Median (min–max)	27.09 (20.08–38.69)	34.7 (29.3–39.6)	
CD8+ (10^3^/mm^3^)	Mean ± SD	0.56 ± 0.24	0.80 ± 0.20	<0.001
	Median (min–max)	0.48 (0.28–1.49)	0.82 (0.44–1.10)	

SD, standard deviation.

**Table 5 ijms-20-03145-t005:** General characteristics of the study and the control group.

Parameter			Study Group (39)	Control Group (20)
**Gender (Number and %)**		**Women**	32 (82.05%)	15 (75%)
		Men	7 (17.95%)	5 (25%)
Age (years)		Mean ± SD	41.49 ± 15.74	42.15 ± 10.38
		Median (min.–max.)	39 (22–95)	40 (29–60)
Duration of hyperthyroidism symptoms (months)		Mean ± SD	2.57 ± 1.91	
		Median (min.–max.)	2 (0–8)	
TSI (U/L)	presence		present	absent
	value	Mean ± SD	12.63 ± 9.41	
		Median (min.–max.)	11.2 (1.5–39.4)	
Anti-TPO U/mL		Median (min.–max.)	1009 (13.7–22810)	11 (5–201)
		Q1–Q3	107.7–2456	8–17.5
Anti-Tg (IU/mL)		Median (min.–max.)	121.5 (10–1360)	12 (10–364)
		Q1–Q3	15–304	10–45
TSH (mIU/L)		Median (min.–max.)	0.008 (0.005–0.03)	1.42 (0.72–2.6)
		Q1–Q3	0.008–0.008	1.22–1.77
FT4 (ng/dL)		Median (min.–max.)	4.78 (2.14–8.02)	
		Q1–Q3	3.94–5.81	
FT3 (pg/mL)		Median (min.–max.)	17.35 (5.6–133)	
		Q1–Q3	11.6–20	
Leukocytes (1 × 10^9^/L)		Mean ± SD	5.98 ± 1.72	6.23 ± 1.34
		Median (min.–max.)	5.66 (3.42–9.99)	5.84 (4.12–9.68)

SD, standard deviation.

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
