# Peer review of "Does the Epstein–Barr Virus Play a Role in the Pathogenesis of Graves’ Disease?"

_ijms, 2019, doi:10.3390/ijms20133145_

Round 1
Reviewer 1 Report
I confirm I am happy with the amendments that have been made and recommend that this paper be published in its current format.
Author Response
Thank you for all the valuable tips and remarks.
Reviewer 2 Report
No significant improvement was done; the manuscript still contains severe flaws and should be rejected. The sample size is too small and conclusions too strong for the study design and methods used in the study. The introduction is written like a review paper, not a research article.
Author Response
Again, we would like to thank you for their valuable suggestions of revisions, which undoubtedly further improved our paper.
We are aware that a small sample size is the main limitation of our study. Still, to the best of our knowledge, our study is the first to analyze possible cross-links between Graves’ disease, EBV infection, and PD-1/PD-L1 expression, and their influence on clinical state of the patients. Strict inclusion criteria applied in our study, such as lack of comorbidities or recent co-infections, significantly restricted the possibility of patient recruitment and affected the size of the study group.
Additionally, to further address this issue, we modified Discussion section indicating small sample size as a main limitation of our study.

Reviewer 3 Report
The authors have improved the manuscript considerably. However, some additional changes are still required. First, it is important that the authors differentiate between somatic and neuropsychiatric symptoms that are the hallmark of Graves' disease.
For example, on lines 28 to 30 the authors note characteristic symptoms that are somatic, but later discuss symptoms that are neuropsychiatric. I suggest changing "Characteristic symptoms" to "Characteristic somatic symptoms" and add the following sentence next: "Characteristic neuropsychiatric symptoms include irritability, anxiety, etc. (based on your own study variables and Stern and colleagues (1996). I would then continue with the differentiation of symptoms in the results and discussion so as to be clear as to what the relationship was between the EBV and the two classes of symptoms.
Finally, on line102 the authors indicate that fatigue wasn't associated with EBV DNA, but then discuss the prevalence of chronic fatigue syndrome in 11 of the 12 GD patients with EBV DNA in lines 228 to 230. How can this be possible? This appears inconsistent.
Author Response
Again, we would like to thank you for their valuable suggestions of revisions, which undoubtedly further improved our paper.
We would like to thank for this important comment. We clarified this inconsistency by dividing the symptoms into somatic and neuropsychiatric, as suggested.
Chronic fatigue was present in 11 out of 12 (92%) patients with EBV DNA, but also in 22 out of 27 (81%) patients without EBV DNA. The difference in chronic fatigue distribution between EBV DNA (-) and EBV DNA (+) groups was not statistically significant (P = 0.42). We removed this confusing paragraph from the Discussion section.

Reviewer 4 Report
In this short, but well written Manuscript, the authors demonstrated the possible involvement of Epstein-Barr Virus (EBV) in the development of Graves disease (GD). The authors found that the circulating mononuclear cells obtained from GD patients possess the higher copy number of EBV DNA, which further correlates with the clinical picture of GD.
In addition to the other reviewers, i have two minor points only:
-how were the PBMCs isolated, the method should be described (Ficoll, etc...)
-was the Blood-Serum used in the other assays? In this study?
Author Response
We would like to thank you for their valuable suggestions of revisions, which undoubtedly further improved our paper.
We added information about PBMCs isolation and Blood-Serum using according to reviewers request. Moreover, the manuscript was checked by the professional native-English medical writer.
Round 2
Reviewer 3 Report
N/A
Author Response
Thank you very much for all your comments. Once again the manuscript was checked by the professional native-English medical writer.
This manuscript is a resubmission of an earlier submission. The following is a list of the peer review reports and author responses from that submission.
Round 1
Reviewer 1 Report
The authors of the present investigation examined the relationship between Epstein-Barr virus infection and the diagnosis of Graves' disease. The result suggest that Graves' disease may be associated with Epstein-Barr infection.
Specific Comments
Page 3, Line 11
This statement will require a citation and reference.
Page 3, Line 27
Replace the phrase "cell immortality". with the phrase "cell immortality."
Page 4, Line 19
What types of analyses were performed and for what purpose? If chi square was used, how were the expected values obtained, etc.
Page 4, Table 1
I suggest the authors recheck their analyses. The reported standard deviations appear to be too high. What does it mean to have a standard deviation that is nearly double the median and potentially the mean (e.g., 74.71 ±123.97)? A quick simulation would produce negative frequencies. What were the mean values?
Replace the term "Medium"with "Median."
Page 5, Line 3
What are "clinical parameters?"
Page 5, Line 23
I suggest replacing "Th" with the letter "T."
Page 6, Line 13
The sentence is missing a comma after the word "GD."
General Comments
Introduction.
Although the non-specific symptoms associated with Graves' disease were examined, no discussion of the history of those symptoms was included in the introduction of the manuscript. Further, the authors failed to put forth a testable hypothesis related to those symptoms.
Results and Conclusions.
While the purpose of the investigation was to examine relationship between Epstein-Barr virus infection and the diagnosis of Graves' disease, the authors also appeared to have examined the non-specific symptoms or "clinical parameters" associated with the disorder. However, measures of the non-specific symptoms were never discussed and the results of those descriptive and inferential analyses were never formally presented. The authors chose to present instead the non-specific symptoms associated with the Graves' patients (Table 3). Table 3 should contain the descriptive statistics associated with Graves' patients and age-matched controls.
Abbreviations.
I suggest the authors reduce the number of acronyms in the manuscript (N = 37) and use only those acronyms that are well accepted by the academic community.
Discussion.
Similar to the general comment made for the introduction, the authors failed to meaningfully discuss the results associated with the non-specific symptoms (e.g., Page 6, Line 44).
Reviewer 2 Report
The authors present a data on a small number of patients with Graves's disease trying to establish a relationship with EBV infection. The sample size is too small given the frequency of GD and the established relationship is pretty weak. Many other factors (confounding factors) may have the impact on EBV presence in humans.
Introduction does not need to contain the pathogenesis of the disease as this is not a review paper. Similarly, there is no need for EBV structure.
The authors do no provide the data on the controls for EBV measurement.
The manuscript requires a proofreading.
Reviewer 3 Report
Overall comments:
· The written English needs attention – some places more so than others (see annotated PDF)
· The data is very light – one table. The author’s also refer to data on lymphocytosis (p5, lines 35-37) but I don’t see this data presented in the manuscript. Including this would certainly help elevate the paper.
· More clarity is needed throughout – e.g. defining acronyms on first use (and correctly! i.e. AIDS not AID or AIDs), giving more detail on the significance of the genetic factors detailed in Figure 1 (RB1 *0304, DQB1 *02…this means nothing to the non-specialist)
· On page 3, they state that the natural host of EBV “is a man” – now I know what they meant by this, but strictly speaking it’s not true! Women get it too. To correct this, they just need to remove the word “a”. It’s points like this that emphasise the need to take care with the written English. I would urge the authors to have the revised manuscript reviewed by a native English speaker to help with this.
· I think a lot of important information has been omitted. For example, how “newly recognized” are the GD patients? Are we talking within the past month? Year? 5 years? Also, is there a difference between EBV copy number (mean, median or medium) between men and women?
· The authors also gloss over the fact that only 30% of the GD patients are EBV seropositive. This makes it seem a little misleading saying that EBV copy number correlates with GD.
· On that latter point, they claim that there is no correlation between EBV incidence and GD signs and symptoms, but I don’t see any concrete data to support this.
